# Distilling Image Classifiers in Object Detectors

**Shuxuan Guo**[1,2*]     **Jose M. Alvarez**[2]     **Mathieu Salzmann**[1]

[1]CVLab, EPFL, Lausanne 1015, Switzerland
[2]NVIDIA, Santa Clara, CA 95051, USA
shuxuan.guo@epfl.ch, josea@nvidia.com, mathieu.salzmann@epfl.ch

## Abstract

Knowledge distillation constitutes a simple yet effective way to improve the performance of a compact student network by exploiting the knowledge of a more powerful teacher. Nevertheless, the knowledge distillation literature remains limited to the scenario where the student and the teacher tackle the same task. Here, we investigate the problem of transferring knowledge not only across architectures but also across tasks. To this end, we study the case of object detection and, instead of following the standard detector-to-detector distillation approach, introduce a classifier-to-detector knowledge transfer framework. In particular, we propose strategies to exploit the classification teacher to improve both the detector's recognition accuracy and localization performance. Our experiments on several detectors with different backbones demonstrate the effectiveness of our approach, allowing us to outperform the state-of-the-art detector-to-detector distillation methods.

## 1   Introduction

Object detection plays a critical role in many real-world applications, such as autonomous driving and video surveillance. While deep learning has achieved tremendous success in this task [25, 26, 31, 32, 40], the speed-accuracy trade-off of the resulting models remains a challenge. This is particularly important for real-time prediction on embedded platforms, whose limited memory and computation power impose strict constraints on the deep network architecture.

To address this, much progress has recently been made to obtain compact deep networks. Existing methods include pruning [1, 2, 13, 22, 38] and quantization [7, 30, 44], both of which aim to reduce the size of an initial deep architecture, as well as knowledge distillation, whose goal is to exploit a deep teacher network to improve the training of a given compact student one. In this paper, we introduce a knowledge distillation approach for object detection.

While early knowledge distillation techniques [18, 33, 36] focused on the task of image classification, several attempts have nonetheless been made for object detection. To this end, existing techniques [5, 12, 39] typically leverage the fact that object detection frameworks consist of three main stages depicted by Figure 1(a): A backbone to extract features; a neck to fuse the extracted features; and heads to predict classes and bounding boxes. Knowledge distillation is then achieved using a teacher with the same architecture as the student but a deeper and wider backbone, such as a Faster RCNN [32] with ResNet152 [14] teacher for a Faster RCNN with ResNet50 student, thus facilitating knowledge transfer at all three stages of the frameworks. To the best of our knowledge, [43] constitutes the only exception to this strategy, demonstrating distillation across different detection frameworks, such as from a RetinaNet [25] teacher to a RepPoints [40] student. This method, however, requires the teacher and the student to rely on a similar detection strategy, i.e., both must be either one-stage detectors or two-stage ones, and, more importantly, still follows a detector-to-detector approach to distillation.

---

[*]The work is done during an internship at NVIDIA.

35th Conference on Neural Information Processing Systems (NeurIPS 2021).

In other words, the study of knowledge distillation remains limited to transfer across architectures tackling the same task. Our classification teacher tackles a different task from the detection student and is trained in a different manner but on the same dataset. Therefore, the classification teacher is capable of providing a different knowledge to the student, for both classification and localization, than that extracted by a detection teacher.

In this paper, we investigate the problem of transferring knowledge not only across architectures but also across tasks. In particular, we observed that the classification head of state-of-the-art object detectors still typically yields inferior performance compared to what can be expected from an image classifier. Thus, as depicted by Figure 1(b), we focus on the scenario where the teacher is an image classifier while the student is an object detector. We then develop distillation strategies to improve both the recognition accuracy and the localization ability of the student.

Our contributions can thus be summarized as follows:

- We introduce the idea of classifier-to-detector knowledge distillation to improve the performance of a student detector using a classification teacher.
- We propose a distillation method to improve the student's classification accuracy, applicable when the student uses either a categorical cross-entropy loss or a binary cross-entropy one.
- We develop a distillation strategy to improve the localization performance of the student be exploiting the feature maps from the classification teacher.

We demonstrate the effectiveness of our approach on the COCO2017 benchmark [23] using diverse detectors, including the relatively large two-stage Faster RCNN and single-stage RetinaNet used in previous knowledge distillation works, as well as more compact detectors, such as SSD300, SSD512 [26] and Faster RCNNs[32] with lightweight backbones. Our classifier-to-detector distillation approach outperforms the detector-to-detector distillation ones in the presence of compact students, and helps to further boost the performance of detector-to-detector distillation techniques for larger ones, such as Faster RCNN and RetinaNet with a ResNet50 backbone. Our code is avlaible at: `https://github.com/NVlabs/DICOD`.

## 2  Related work

**Object detection** is one of the fundamental tasks in computer vision, aiming to localize the objects observed in an image and classify them. Recently, much progress has been made via the development of both one-stage [9, 21, 26, 31, 37] and two-stage [4, 15, 24, 32] deep object detection frameworks, significantly improving the mean average precision (mAP) on standard benchmarks [10, 11, 23]. However, the performance of these models typically increases with their size, and so does their inference runtime. This conflicts with their deployment on embedded platforms, such as mobile phones, drones, and autonomous vehicles, which involve computation and memory constraints. While some efforts have been made to design smaller detectors, such as SSD [26], YOLO [31] and detectors with lightweight backbones [19, 35], the performance of these methods does not match that of deeper ones.

**Knowledge distillation** offers the promise to boost the performance of such compact networks by exploiting deeper teacher architectures. Early work in this space focused on the task of image classification. In particular, Hinton et al. [18] proposed to distill the teacher's class probability distribution into the student, and Romero et al. [33] encouraged the student's intermediate feature maps to mimic the teacher's ones. These initial works were followed by a rapid growth in the number of knowledge distillation strategies, including methods based on attention maps [42], on transferring feature flows defined by the inner product of features [41], and on contrastive learning to structure the knowledge distilled from teacher to the student [36]. Heo et al. [17] proposed a synergistic distillation strategy aiming to jointly leverage a teacher feature transform, a student feature transform, the distillation feature position and a better distance function.

Compared to image classification, object detection poses the challenge of involving both recognition and localization. As such, several works have introduced knowledge distillation methods specifically tailored to this task. This trend was initiated by Chen et al. [5], which proposed to distill knowledge from a teacher detector to a student detector in both the backbone and head stages. Then, Wang et al. [39] proposed to restrict the teacher-student feature imitation to regions around positive anchor

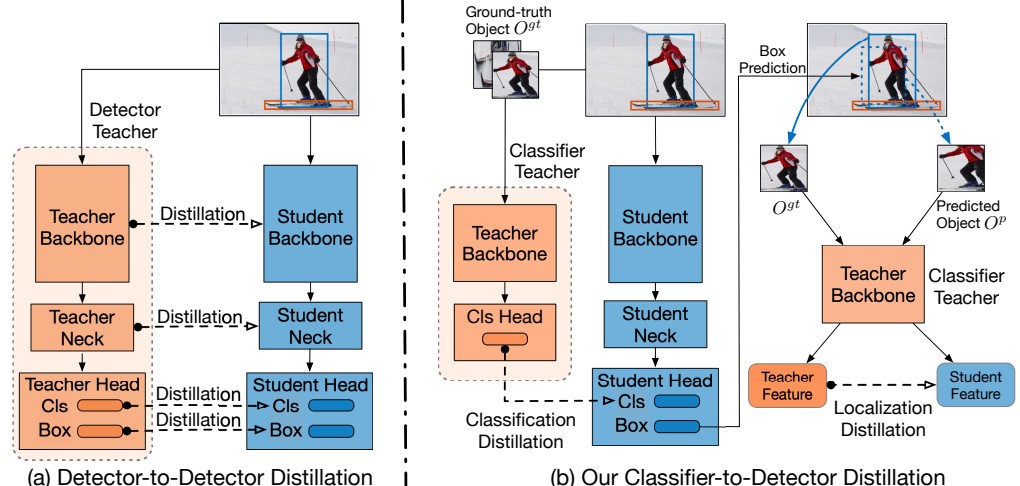

Figure 1: **Overview of our classifier-to-detector distillation framework.** (a) Existing methods perform distillation across corresponding stages in the teacher and student, which restricts their applicability to detector-to-detector distillation. (b) By contrast, we introduce strategies to transfer the knowledge from an image classification teacher to an object detection student, improving both its recognition and localization accuracy.

boxes; Dai et al. [8] produced general instances based on both the teacher's and student's outputs, and distilled feature-based, relation-based and response-based knowledge in these general instances; Guo et al. [12] proposed to decouple the intermediate features and classification predictions of the positive and negative regions during knowledge distillation. All the aforementioned knowledge distillation methods require the student and the teacher to follow the same kind of detection framework, and thus typically transfer knowledge between models that only differ in terms of backbone, such as from a RetinaNet-ResNet152 to a RetinaNet-ResNet50. In [43], such a constraint was relaxed via a method able to transfer knowledge across the feature maps of different frameworks. This allowed the authors to leverage the best one-stage, resp. two-stage, teacher model to perform distillation to any one-stage, resp. two-stage, student. This method, however, still assumes that the teacher is a detector.

In short, existing knowledge distillation methods for object detection all follow a detector-to-detector transfer strategy. In fact, to the best of our knowledge, distillation has only been studied across two architectures that tackle the same task, may it be image classification, object detection, or even semantic segmentation [16, 27]. In this paper, by contrast, we investigate the use of knowledge distillation across tasks and develop strategies to distill the knowledge of an image classification teacher to an object detection student.

## 3 Our Approach

Our goal is to investigate the transfer of knowledge from an image classifier to an object detector. As illustrated in Figure 1, this contrasts with existing knowledge distillation techniques for object detection, which typically assume that the teacher and the student both follow a similar three-stage detection pipeline. For our classifier-to-detector knowledge distillation to be effective, we nonetheless need the student and teacher to process the same data and use the same loss for classification. To this end, given a detection dataset $\mathcal{D}_{det}$ depicting $C$ foreground object categories, we construct a classification dataset $\mathcal{D}_{cls}$ by extracting all objects from $\mathcal{D}_{det}$ according to their ground-truth bounding boxes and labels. We then train our classification teacher $\mathcal{F}_t$, with parameters $\theta^t$, on $\mathcal{D}_{cls}$ in a standard classification manner. In the remainder of this section, we introduce our strategies to exploit the resulting teacher to improve both the classification and localization accuracy of the student detector $\mathcal{F}_s$, with parameters $\theta^s$.

### 3.1 KD$_{cls}$: Knowledge Distillation for Classification

Our first approach to classifier-to-detector distillation focuses on the classification accuracy of the student network. To this end, we make use of the class-wise probability distributions obtained by the teacher and the student, softened by making use of a temperature parameter $T$. Below, we first derive our general formulation for distillation for classification, and then discuss in more detail how we obtain the teacher and student class distributions for the two types of classification losses commonly used by object detection frameworks.

Formally, given $K$ positive anchor boxes or object proposals, which are assigned with one of the ground-truth labels and bounding boxes during training, let $p_k^{s,T}$ denote the vector of softened class probabilities for box $k$ from the student network, obtained at temperature $T$, and let $p_k^{t,T}$ denote the corresponding softened probability vector from the teacher network. We express knowledge distillation for classification as a loss function measuring the Kullback-Leibler (KL) divergence between the teacher and student softened distributions. This can be written as

$$\mathcal{L}_{kd-cls} = \frac{1}{K} \sum_{k=1}^{K} KL(p_k^{t,T} \parallel p_k^{s,T}) \ . \tag{1}$$

The specific way we define the probability vectors $p_k^{s,T}$ and $p_k^{t,T}$ then depends on the loss function that the student detector uses for classification. Indeed, existing detectors follow two main trends: some, such as Faster RCNN and SSD, exploit the categorical cross-entropy loss with a softmax, accouting for the $C$ foreground classes and 1 background one; others, such as RetinaNet, employ a form of binary cross-entropy loss with a sigmoid[2], focusing only on the $C$ foreground classes. Let us now discuss these two cases in more detail.

**Categorical cross-entropy.** In this case, for each positive object bounding box $k$, the student detector outputs logits $z_k^s \in (C+1)$. We then compute the corresponding softened probability for class $c$ with temperature $T$ as

$$p_k^{s,T}(c|\theta^s) = \frac{e^{z_{k,c}^s/T}}{\sum_{j=1}^{C+1} e^{z_{k,j}^s/T}} \ , \tag{2}$$

where $z_{k,c}^s$ denote the logit corresponding to class $c$. By contrast, as our teacher is a $C$-way classifier, it produces logits $z_k^t \in C$. We thus compute its softened probability for class $c$ as

$$\tilde{p}_k^{t,T}(c|\theta^t) = \frac{e^{z_{k,c}^t/T}}{\sum_{j=1}^{C} e^{z_{k,j}^t/T}} \ , \tag{3}$$

and, assuming that all true objects should be classified as background with 0 probability, augment the resulting distribution to account for the background class as $p^{t,T} = [\tilde{p}^{t,T}, 0]$.

The KL-divergence between the teacher and student softened distributions for object $k$ can then be written as

$$KL(p_k^{t,T} \parallel p_k^{s,T}) = T^2 \sum_{c=1}^{C+1} p_{k,c}^{t,T} \log p_{k,c}^{t,T} - p_{k,c}^{t,T} \log p_{k,c}^{s,T} \ . \tag{4}$$

**Binary cross-entropy.** The detectors that rely on the binary cross-entropy output a score between 0 and 1 for each of the $C$ foreground classes, but, together, these scores do not form a valid distribution over the $C$ classes as they do not sum to 1. To nonetheless use them in a KL-divergence measure between the teacher and student, we rely on the following strategy. Given the student and teacher $C$-dimensional logit vectors for an object $k$, we compute softened probabilities as

$$\tilde{p}_k^{s,T}(c|\theta^s) = (1 + e^{-z_{k,c}^s/T})^{-1} \ ,$$
$$\tilde{p}_k^{t,T}(c|\theta^t) = (1 + e^{-z_{k,c}^t/T})^{-1} \ . \tag{5}$$

We then build a 2-class (False-True) probability distribution for each category according to the ground-truth label $l$ of object $k$. Specifically, for each category $c$, we write

$$p_{k,c}^{s,T} = [1 - \tilde{p}_{k,c}^{s,T}, \tilde{p}_{k,c}^{s,T}], \tag{6}$$

---

[2]In essence, the RetinaNet focal loss follows a binary cross-entropy formulation.

for the student, and similarly for the teacher. This lets us express the KL-divergence for object $k$ as

$$KL(p_k^{t,T} \parallel p_k^{s,T}) = \frac{T^2}{C} \sum_{c=1}^{C} \sum_{i=0}^{1} p_{k,c}^{t,T}(i) \log p_{k,c}^{t,T}(i) - p_{k,c}^{t,T}(i) \log p_{k,c}^{s,T}(i) , \tag{7}$$

where $p_{k,c}^{t,T}(i)$ indicates the $i$-th element of the 2-class distribution $p_{k,c}^{t,T}$.

## 3.2  $\text{KD}_{loc}$: Knowledge Distillation for Localization

While, as will be shown by our experiments, knowledge distillation for classification already helps the student detector, it does not aim to improve its localization performance. Nevertheless, localization, or bounding box regression, is critical for the success of a detector and is typically addressed by existing detector-to-detector distillation frameworks [5, 8]. To also tackle this in our classifier-to-detector approach, we develop a feature-level distillation strategy, exploiting the intuition that the intermediate features extracted by the classification teacher from a bounding box produced by the student should match those of the ground-truth bounding box.

Formally, given an input image $I$ of size $w \times h$, let us denote by $B_k = (x_1, y_1, x_2, y_2)$ the top-left and bottom-right corners of the $k$-th bounding box produced by the student network. Typically, this is achieved by regressing the offset of an anchor box or object proposal. We then make use of a Spatial Transformer [20] unit to extract the image region corresponding to $B_k$. It is a non-parametric differentiable module that links the regressed bounding boxes with the classification teacher to yield an end-to-end model during training. Specifically, we compute the transformer matrix

$$A_k = \begin{bmatrix} (x_2 - x_1)/w & 0 & -1 + (x_1 + x_2)/w \\ 0 & (y_2 - y_1)/h & -1 + (y_1 + y_2)/h \end{bmatrix} , \tag{8}$$

which allows us to extract the predicted object region $O_k^p$ with a grid sampling size $s$ as

$$O_k^p = f_{ST}(A_k, I, s) , \tag{9}$$

where $f_{ST}$ denotes the spatial transformer function. As illustrated in the right portion of Figure 1(b), we then perform distillation by comparing the teacher's intermediate features within the predicted object region $O_k^p$ to those within its assigned ground-truth one $O_k^{gt}$.

Specifically, for a given layer $\ell$, we seek to compare the features $\mathcal{F}_t^\ell(O_k^p)$ and $\mathcal{F}_t^\ell(O_k^{gt})$ of the positive box $k$. To relax the pixel-wise difference between the features, we make use of the adaptive pooling strategy of [28], which produces a feature map $AP(\mathcal{F}_t^\ell(O))$ of a fixed size $M \times W \times H$ from the features extracted within region $O$. We therefore write our localization distillation loss as

$$\mathcal{L}_{kd-loc}^L = \frac{1}{KLMHW} \sum_{k=1}^{K} \sum_{\ell=1}^{L} \mathbb{1}_\ell \|AP(\mathcal{F}_t^\ell(O_k^p)) - AP(\mathcal{F}_t^\ell(O_k^{gt}))\|_1 , \tag{10}$$

where $K$ is the number of positive anchor boxes or proposals, $L$ is the number of layers at which we perform distillation, $\mathbb{1}_l$ is the indicator function to denote whether the layer $\ell$ is used or not to distill knowledge, and $\| \cdot \|_1$ denotes the $L_1$ norm. As both the spatial transformer and the adaptive pooling operation are differentiable, this loss can be backpropagated through the student detector.

Note that, as a special case, our localization distillation strategy can be employed not only on intermediate feature maps but on the object region itself ($\ell_0$), encouraging the student to produce bounding boxes whose underlying image pixels match those of the ground-truth box. This translates to a loss function that does not exploit the teacher and can be expressed as

$$\mathcal{L}_{kd-loc}^0(O^p, O^{gt}) = \frac{1}{KMHW} \sum_{k=1}^{K} \|AP(O_k^p) - AP(O_k^{gt})\|_1 . \tag{11}$$

Depending on the output size of the adaptive pooling operation, this loss function encodes a more-or-less relaxed localization error. As will be shown by our experiments, it can serve as an attractive complement to the standard bounding box regression loss of existing object detectors, whether using distillation or not.

Table 1: **Analysis of our classifier-to-detector distillation method with compact students on the COCO2017 validation set.** R50 is ResNet50, MV2 is MobileNetV2, QR50 is quartered ResNet50.

| Method | mAP | $AP_{50}$ | $AP_{75}$ | $AP_s$ | $AP_m$ | $AP_l$ | mAR | $AR_s$ | $AR_m$ | $AR_l$ |
|---|---|---|---|---|---|---|---|---|---|---|
| SSD300-VGG16 | 25.6 | 43.8 | 26.3 | 6.8 | 27.8 | 42.2 | 37.6 | 12.5 | 41.7 | 58.6 |
| + $KD_{cls}$ | 26.3 (↑ 0.7) | 45.2 | 27.2 | 7.3 | 28.5 | 43.6 | 38.4 | 12.8 | 42.6 | 59.1 |
| + $KD_{loc}^0$ | 27.1 (↑ 1.5) | 43.2 | 28.4 | 7.5 | 29.4 | 43.3 | 40.0 | 13.4 | 44.4 | 60.6 |
| + $KD_{loc}$ | 27.2 (↑ 1.6) | 43.3 | 28.5 | 7.5 | 29.5 | 43.5 | 40.2 | 13.2 | 44.7 | 61.5 |
| + $KD_{cls}$ + $KD_{loc}$ | **27.9 (↑ 2.3)** | **45.1** | **29.2** | **8.1** | **30.1** | **45.4** | **40.4** | **13.9** | **44.7** | **61.4** |
| SSD512-VGG16 | 29.4 | 49.3 | 31.0 | 11.7 | 34.1 | 44.9 | 42.7 | 17.6 | 48.7 | 60.6 |
| + $KD_{cls}$ | 30.3 (↑ 0.9) | 51.1 | 31.7 | 12.7 | 34.6 | 45.5 | 43.3 | 19.4 | 49.0 | 60.4 |
| + $KD_{loc}^0$ | 30.8 (↑ 1.4) | 48.8 | 32.9 | 12.8 | 35.8 | 46.2 | 44.7 | 18.8 | 51.1 | 63.4 |
| + $KD_{loc}$ | 31.0 (↑ 1.6) | 49.1 | 32.8 | 12.6 | 35.8 | 46.2 | 45.0 | 18.9 | 51.6 | 63.2 |
| + $KD_{cls}$ + $KD_{loc}$ | **32.1 (↑ 2.7)** | **51.0** | **34.0** | **13.3** | **36.6** | **47.9** | **45.3** | **20.1** | **51.2** | **63.1** |
| Faster RCNN-QR50 | 23.3 | 40.7 | 23.9 | 13.1 | 25.0 | 30.7 | 40.2 | 22.7 | 42.8 | 51.8 |
| + $KD_{cls}$ | 25.9 (↑ 2.6) | 45.5 | 26.2 | 15.3 | 27.9 | 34.0 | 42.8 | 25.5 | 46.0 | 54.9 |
| + $KD_{loc}^0$ | 24.2 (↑ 0.9) | 41.1 | 25.0 | 13.7 | 25.8 | 32.1 | 41.7 | 23.8 | 44.3 | 54.8 |
| + $KD_{loc}$ | 24.3 (↑ 1.0) | 41.0 | 25.1 | 13.0 | 25.9 | 32.5 | 41.6 | 22.7 | 44.6 | 54.7 |
| + $KD_{cls}$ + $KD_{loc}$ | **27.2 (↑ 3.9)** | **46.0** | **27.7** | **15.2** | **29.3** | **36.2** | **44.5** | **25.9** | **48.1** | **58.3** |
| Faster RCNN-MV2 | 31.9 | 52.0 | 34.0 | 18.5 | 34.4 | 41.0 | 47.5 | 29.7 | 50.9 | 60.4 |
| + $KD_{cls}$ | 32.6 (↑ 0.7) | 53.3 | 34.6 | 18.9 | 34.8 | 42.3 | 48.1 | 29.7 | 51.2 | 61.5 |
| + $KD_{loc}^0$ | 32.2 (↑ 0.3) | 51.9 | 34.2 | 18.3 | 34.4 | 41.8 | 47.9 | 29.0 | 50.8 | 61.5 |
| + $KD_{loc}$ | 32.3 (↑ 0.4) | 52.0 | 34.7 | 18.1 | 34.8 | 41.6 | 48.0 | 28.7 | 51.3 | 61.6 |
| + $KD_{cls}$ + $KD_{loc}$ | **32.7 (↑ 0.8)** | **52.9** | **35.0** | **19.0** | **35.0** | **42.9** | **48.4** | **29.9** | **51.8** | **61.9** |

## 3.3 Overall Training Loss

To train the student detector given the image classification teacher, we then seek to minimize the overall loss

$$\mathcal{L} = \mathcal{L}_{det} + \lambda_{kc}\mathcal{L}_{kd-cls} + \lambda_{kl}\mathcal{L}_{kd-loc} , \tag{12}$$

where $\mathcal{L}_{det}$ encompasses the standard classification and localization losses used to train the student detector of interest. $\lambda_{kc}$ and $\lambda_{kl}$ are hyper-parameters setting the influence of each loss.

## 4 Experiments

In this section, we first conduct a full study of our classification and localization distillation methods on several compact detectors, and then compare our classifier-to-detector approach to the state-of-the-art detector-to-detector ones. Finally, we perform an extensive ablation study of our method and analyze how it improves the class recognition and localization in object detection. All models are trained and evaluated on MS COCO2017 [23], which contains over 118k images for training and 5k images for validation (minival) depicting 80 foreground object classes. Our implementation is based on MMDetection [6] with Pytorch [29]. Otherwise specified, we take the ResNet50 as the classification teacher. We will use the same teacher for all two-stage Faster RCNNs and one-stage RetinaNets in our classifier-to-detector distillation method. We consider this to be an advantage of our method, since it lets us use the same teacher for multiple detectors. To train this classification teacher, we use the losses from Faster RCNN and RetinaNet frameworks jointly. Since SSDs use different data augmentation, we train another ResNet50 classification teacher for them. Additional experimental details on how to train our classification teachers are provided in the supplementary material.

### 4.1 Classifier-to-Detector Distillation on Compact Students

We first demonstrate the effectiveness of our classifier-to-detector distillation method on compact detectors, namely, SSD300, SSD512 [26] and the two-stage Faster RCNN [32] detector with lightweight backbones, i.e., MobileNetV2 [35] and Quartered-ResNet50 (QR50), obtained by dividing the number of channels by 4 in every layer of ResNet50, reaching a 66.33% top-1 accuracy on ImageNet [34].

**Experimental setting.** All object detectors are trained in their default settings on Tesla V100 GPUs. The SSDs follows the basic training recipe in MMDetection [6]. The lightweight Faster RCNNs are

trained with a $1\times$ training schedule for 12 epochs. The details for the training settings of each model are provided in the supplementary material. We use a ResNet50 with input resolution $112 \times 112$ as classification teacher for all student detectors. We report the mean average precision (mAP) and mean average recall (mAR) for intersection over unions (IoUs) in [0.5:0.95], the APs at IoU=0.5 and 0.75, and the APs and ARs for small, medium and large objects.

**Results.** The results are shown in Table 1. Our classification distillation yields improvements of at least 0.7 mAP for all student detectors. It reaches a 2.6 mAP improvement for Faster RCNN-QR50, which indicates that the classification in this model is much weaker. The classification distillation improves $AP_{50}$ more than $AP_{75}$, while the localization distillation improves $AP_{75}$ more than $AP_{50}$. As increasing $AP_{75}$ requires more precise localization, these results indicate that each of our distillation losses plays its expected role. Note that the SSDs benefit more from the localization method than the Faster RCNNs. We conjecture this to be due to the denser, more accurate proposals of the Faster RCNNs compared to the generic anchors of the SSDs. Note also that a Faster RCNNs with a smaller backbone benefits more from our distillation than a larger one.

## 4.2 Comparison with Detector-to-detector Distillation

We then compare our classifier-to-detector distillation approach with the state-of-the-art detector-to-detector ones, such as KD [5], FGFI [39], GID [8] and FKD [43]. Here, in addition to the compact students used in Section 4.1, we also report results on the larger students that are commonly used in the literature, i.e., Faster RCNN and RetinaNet with deeper ResNet50 (R50) backbones.

**Experimental setting.** Following [43], the Faster RCNN-R50 and RetinaNet-R50 are trained with a $2\times$ schedule for 24 epochs. To illustrate the generality of our approach, we also report the results of our distillation strategy used in conjunction with FKD [43], one of the current best detector-to-detector distillation methods. Note that, while preparing this work, we also noticed the concurrent work of [12], whose DeFeat method also follows a detector-to-detector distillation approach, and thus could also be complemented with out strategy.

**Results.** We report the results in Table 2. For compact student detectors, such as Faster RCNN-QR50 and SSD512, our classifier-to-detector distillation surpasses the best detector-to-detector one by 1.1 and 0.9 mAP points, respectively. For student detectors with deeper backbones, our method improves the baseline by 0.8, 0.4 and 0.5 points. Furthermore, using it in conjunction with the FKD detector-to-detector distillation method boosts the performance to the state-of-the-art of 28.0, 32.6, 34.2, 41.9 and 40.7 mAP. Overall, these results evidence that our approach is orthogonal to the detector-to-detector distillation methods, allowing us to achieve state-of-the-art performance by itself or by combining it with a detector-to-detector distillation strategy.

Table 2: Comparison to detector-to-detector distillation methods on the COCO2017 validation set.

| Method | mAP | $AP_s$ | $AP_m$ | $AP_l$ |
|---|---|---|---|---|
| Faster RCNN-QR50 | 23.3 | 13.1 | 25.0 | 30.7 |
| + FKD [43] | 26.1 | 14.6 | 27.3 | 35.0 |
| + Ours | 27.2 | 15.2 | 29.3 | 36.2 |
| + Ours + FKD | **28.0** | **15.4** | **29.8** | **38.5** |
| SSD512-VGG16 | 29.4 | 11.7 | 34.1 | 44.9 |
| + FKD [43] | 31.2 | 12.6 | 37.4 | 46.2 |
| + Ours | 32.1 | 13.3 | 36.6 | 47.9 |
| + Ours + FKD | **32.6** | **13.5** | **37.6** | **48.3** |
| Faster RCNN-MV2 | 31.9 | 18.5 | 34.4 | 41.0 |
| + FKD [43] | 33.9 | 18.3 | 36.3 | 45.4 |
| + Ours | 32.7 | **19.0** | 35.0 | 42.9 |
| + Ours + FKD | **34.2** | 18.5 | **36.3** | 45.9 |
| Faster RCNN-R50 | 38.4 | 21.5 | 42.1 | 50.3 |
| + KD [5] | 38.7 | 22.0 | 41.9 | 51.0 |
| + FGFI [39] | 39.1 | 22.2 | 42.9 | 51.1 |
| + GID [8] | 40.2 | 22.7 | 44.0 | 53.2 |
| + FKD [43] | 41.5 | 23.5 | 45.0 | 55.3 |
| + Ours | 38.8 | 22.5 | 42.5 | 50.8 |
| + Ours + FKD | **41.9** | **23.8** | **45.2** | **56.0** |
| RetinaNet-R50 | 37.4 | 20.0 | 40.7 | 49.7 |
| + FGFI [39] | 38.6 | 21.4 | 42.5 | 51.5 |
| + GID [8] | 39.1 | 22.8 | 43.1 | 52.3 |
| + FKD [43] | 39.6 | 22.7 | 43.3 | 52.5 |
| + Ours | 37.9 | 20.5 | 41.3 | 50.5 |
| + Ours +FKD | **40.7** | **23.1** | **44.7** | **53.8** |

## 4.3 Ablation Study

In this section, we investigate the influence of the hyper-parameters and of different classification teachers in our approach. To this end, we use the SSD300 student detector.

**Ablation study of $KD_{cls}$.** We first study the effect of the loss weight $\lambda_{kc}$ and the temperature $T$ for classification distillation. As shown in Table 3a, these two hyper-parameters have a mild impact on the results, and we obtain the best results with $\lambda_{kc} = 0.4$ and $T = 2$, which were used for all other experiments with SSDs.

Table 3: **Ablation study of KD$_{cls}$.** We evaluate the impact of the hyper-parameters and of various classification teachers on our classification distillation.

(a) Varying $\lambda_{kc}$ and $T$.

| $\lambda_{kc}$ | $T$ | mAP | AP$_{50}$ | AP$_{75}$ |
|---|---|---|---|---|
| baseline | / | 25.6 | 43.8 | 26.3 |
| 0.1 | 1 | 25.8 | 44.2 | 26.6 |
| 0.1 | 2 | 25.4 | 44.4 | 25.7 |
| 0.2 | 1 | 25.8 | 44.2 | 26.6 |
| 0.3 | 1 | 26.0 | 44.6 | 26.7 |
| 0.4 | 1 | 26.1 | 44.8 | 26.6 |
| 0.4 | 2 | **26.3** | **45.2** | **27.2** |
| 0.4 | 3 | 26.0 | 45.2 | 26.7 |

(b) Varying the teacher network.

| Teacher | Top-1 | mAP | AP$_{50}$ | AP$_{75}$ |
|---|---|---|---|---|
| ResNet18 | 75.78 | 25.9 | 44.4 | 26.4 |
| ResNet50 | 80.30 | **26.3** | **45.2** | **27.2** |
| ResNeXt101 | 83.35 | 25.3 | 43.3 | 25.8 |

| Input size | Top-1 | mAP | AP$_{50}$ | AP$_{75}$ |
|---|---|---|---|---|
| $56 \times 56$ | 76.26 | 26.2 | 44.8 | 26.9 |
| $112 \times 112$ | 80.30 | **26.3** | **45.2** | **27.2** |
| $224 \times 224$ | 80.41 | 26.2 | 44.9 | 26.9 |

Table 4: **Ablation study of KD$_{loc}$.** We investigate the effect of the sampling size, the pooling size and the choice of distilled layers on our localization distillation.

(a) Varying the sampling size.

| Sampling size | mAP | AP$_{50}$ | AP$_{75}$ |
|---|---|---|---|
| $14 \times 14$ | 26.4 | 43.0 | 27.0 |
| $28 \times 28$ | 26.7 | 43.2 | 27.8 |
| $56 \times 56$ | 26.8 | 43.3 | 28.0 |
| $112 \times 112$ | **27.0** | 43.5 | 28.1 |
| $224 \times 224$ | **27.0** | 43.4 | 28.2 |

(b) Varying the pooling size.

| Pooling size | mAP | AP$_{50}$ | AP$_{75}$ |
|---|---|---|---|
| $2 \times 2$ | 26.6 | 43.5 | 27.5 |
| $4 \times 4$ | 27.0 | 43.5 | 28.1 |
| $8 \times 8$ | **27.1** | 43.2 | 28.4 |
| $16 \times 16$ | 26.9 | 42.8 | 28.1 |

(c) Varying distilled layers.

| $\ell_0$ | $\ell_1$ | $\ell_2$ | mAP |
|---|---|---|---|
| ✓ | | | 27.1 |
| | ✓ | | 26.8 |
| ✓ | ✓ | | **27.2** |
| ✓ | ✓ | ✓ | 26.9 |

We then investigate the impact of different classification teacher networks. To this end, we trained three teacher networks ranging from shallow to deep: ResNet18, ResNet50 and ResNext101-32×8d. We further study the impact of the input size to these teachers on classification distillation, using the three sizes $[56 \times 56, 112 \times 112, 224 \times 224]$. As shown in Table 3b, even the shallow ResNet18 classification teacher can improve the performance of the student detector by 0.3 points, and the improvement increases by another 0.4 points with the deeper ResNet50 teacher. However, the performance drops with the ResNeXt101 teacher, which is the teacher with the highest top-1 accuracy. This indicates that a deeper teacher is not always helpful, as it might be overconfident to bring much additional information compared to the ground-truth labels. As for the input size, we observe only small variations across the different sizes, and thus use a size of 112 in all other experiments.

**Ablation study of KD$_{loc}$.** We then evaluate the influence of the two main hyper-parameters of localization distillation, i.e., the grid sampling size of the spatial transformer and the adaptive pooling size of the feature maps. To this end, we vary the sampling size in $[14, 28, 56, 112, 224]$ and the pooling size in $[2 \times 2, 4 \times 4, 8 \times 8, 16 \times 16]$.

As shown in Table 4a, our localization distillation method benefits from a larger sampling size, although the improvement saturates after a size of 112. This lets us use the same classification teacher, with input size 112, for both classification and localization distillation. The adaptive pooling size has a milder effect on the performance, as shown in Table 4b, with a size of 8 yielding the best mAP. In our experiments, we adopt either 4 or 8, according to the best performance on the validation set.

We further study the layers to be distilled in our localization distillation. To this end, we extract features from the first convolutional layer $\ell_1$, and from the following bottleneck block $\ell_2$ of the ResNet50 teacher. As shown in Table 4c, distilling the knowledge of only the object regions ($\ell_0$) yields a better mAP than using the $\ell_1$ features. However, combining the object regions ($\ell_0$) with the feature maps from $\ell_1$ improves the results. Adding more layers does not help, which we conjecture to be due to the fact that these layers extract higher-level features that are thus less localized.

### 4.4 Analysis

To further understand how our classifier-to-detector distillation method affects the quality of the classification and localization, in Table 5, we report the APs obtained with IoUs in $[0.5, 0.95]$ with a step of 0.05. These results highlight that our classification and localization distillation strategies behave differently for different IoU thresholds. Specifically, KD$_{cls}$ yields larger improvements

Table 5: APs for IoUs ranging from 0.5 to 0.95 on the COCO2017 validation set.

| Method | mAP | $AP_{50}$ | $AP_{55}$ | $AP_{60}$ | $AP_{65}$ | $AP_{70}$ | $AP_{75}$ | $AP_{80}$ | $AP_{85}$ | $AP_{90}$ | $AP_{95}$ |
|---|---|---|---|---|---|---|---|---|---|---|---|
| SSD300 | 25.6 | 43.8 | 41.3 | 38.4 | 35.1 | 31.2 | 26.3 | 20.3 | 13.0 | 5.2 | 0.5 |
| + $KD_{cls}$ | 26.3 | 45.2 | 42.6 | 39.9 | 36.1 | 31.6 | 27.2 | 21.0 | 13.5 | 5.1 | 0.5 |
| + $KD_{loc}$ | 27.2 | 43.3 | 41.3 | 38.8 | 36.0 | 32.9 | 28.5 | 23.0 | 16.5 | 8.4 | 1.3 |
| + $KD_{cls}$ + $KD_{loc}$ | 27.9 | 45.1 | 42.8 | 40.2 | 37.0 | 34.0 | 29.2 | 23.9 | 17.0 | 8.8 | 1.2 |

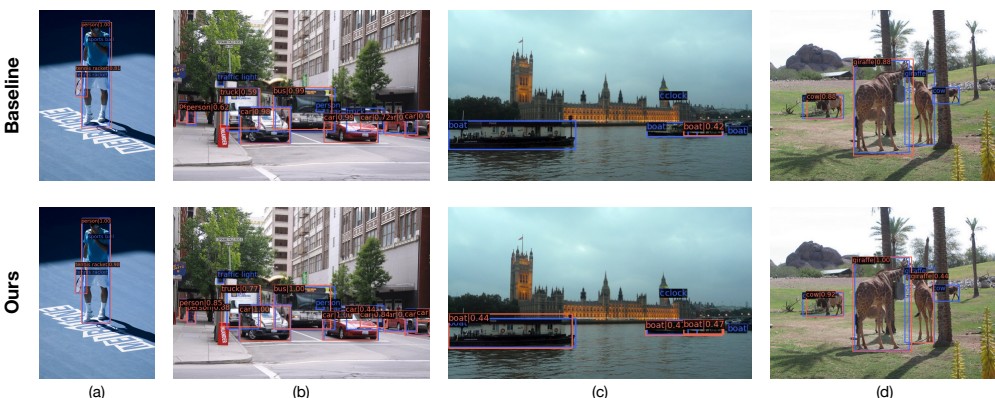

(a)  (b)  (c)  (d)

Figure 3: **Qualitative analysis (better viewed in color).** The ground-truth bounding boxes are in blue with their labels, and the predictions are in red with predicted labels and confidence.

for smaller IoUs, whereas $KD_{loc}$ is more effective with IoUs larger than 0.75. This indicates that $KD_{loc}$ indeed focuses on precise localization, while $KD_{cls}$ distills category information. The complementarity of both terms is further evidenced by the fact that all APs increase when using both of them jointly.

**Detection error analysis.** We analyze the different types of detection errors using the tool proposed by Bolya et al. [3] for the baseline SSD300 and the distilled models with our $KD_{cls}$ and $KD_{loc}$. We focus on the classification and localization errors, which are the main errors in object detection. The details of all error types are provided in the supplementary material. As shown in Figure 2a, $KD_{cls}$ decreases the classification error especially for IoUs smaller than 0.65. By contrast, as shown in Figure 2b, the effect of $KD_{loc}$ increases with the IoU. This again shows the complementary nature of these terms.

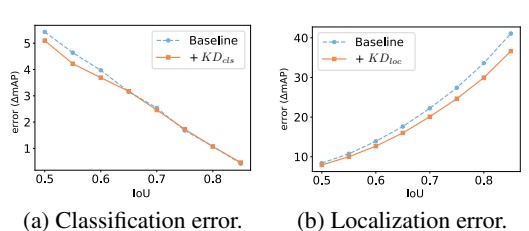

(a) Classification error.  (b) Localization error.

Figure 2: Detection error analysis.

**Qualitative analysis.** Figure 3 compares the detection results of the baseline model and of our distilled model on a few images. We observe that (i) the bounding box predictions of the distilled model are more precise than those of the baseline; (ii) the distilled model generates higher confidences for the correct predictions and is thus able to detect objects that were missed by the baseline, such as the boat in Figure 3c and the giraffe in Figure 3d.

## 5 Conclusion

We have introduced a novel approach to knowledge distillation for object detection, replacing the standard detector-to-detector strategy with a classifier-to-detector one. To this end, we have developed a classification distillation loss function and a localization distillation one, allowing us to exploit the classification teacher in two complementary manners. Our approach outperforms the state-of-the-art detector-to-detector ones on compact student detectors. While the improvement decreases for larger student networks, our approach can nonetheless boost the performance of detector-to-detector distillation. We have further shown that the same classification teacher could be used for all student detectors if they employ the same data augmentation strategy, thus reducing the burden of training a separate teacher for every student detector. Ultimately, we believe that our work opens the door to

a new approach to distillation beyond object detection: Knowledge should be transferred not only across architectures, but also across tasks.

## Broader impact

Knowledge distillation is a simple yet effective method to improve the performance of a compact neural network by exploiting the knowledge of a more powerful teacher model. Our work introduces a general approach to knowledge distillation for object detection to transfer knowledge across architectures and tasks. Our approach enables distilling knowledge from a single classification teacher into different student detectors. As such, our work reduces the need for a separate deep teacher detector for each student networks; therefore, we reduce training resources and memory footprint. As we focus on compact networks, our work could significantly impact applications in resource-constrained environments, such as mobile phones, drones, or autonomous vehicles. We do not foresee any obvious undesirable ethical/social impact at this moment.

## Acknowledgments and Disclosure of Funding

This work was supported in part by the Swiss National Science Foundation and by NVIDIA during an internship. We would like to thank Maying Shen at NVIDIA for the valuable discussions and for running some of the experiments. We also thank NVIDIA for providing their excellent experiment platform and computing resources.

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
