# Distilling Image Classifiers in Object Detectors
## - Supplementary Material -

## S1 Code for Our Approach

As mentioned in Checklist 3(a) and 4(c), we provide the URL (DICOD[1]) for our code and pre-trained classification teacher model to reproduce the main experimental results. The details of how to build up the environment and run our main experiments are in the README.md file.

As mentioned in Checklist 4(a) and 4(b), below, we provide the details and licenses of the existing assets we used in our work, such as the MS COCO2017 [23] dataset and the MMDetection [6] codebase. Both of them are open source and available for non-commercial academic research.

**MS COCO2017** [23][2] is a large-scale object detection, segmentation, key-point detection and captioning dataset. We use its detection benchmark, which consists of 118k training images and 5k validation ones, depicting 80 foreground object classes. The annotations for object detection are bounding boxes and object labels. In our work, we respect the terms of use listed on the website. The annotations in this dataset, along with their website, belong to the COCO Consortium and are licensed under a Creative Commons Attribution 4.0 License.

**MMDetection** [6][3] is an open source object detection toolbox based on Pytorch [29], which is released under the Apache 2.0 license. Together with MMDetection, we also use the MMCV library[4], which is a dependent library for MMDetection. MMCV is mainly released under the Apache 2.0 license, while some specific operations in this library fall under other licenses. Please refer to LICENSES.md in their website.

## S2 Training Classification Teachers

In this section, we provide the details of our experimental classification setup and of training classification teachers.

**Experimental setup**. To train and validate our classification teachers, we use the MS COCO2017 [23] detection dataset and crop all the objects according to their ground-truth bounding boxes. The resulting classification dataset consists of 849,902 objects for training and 36,334 objects for validation. We then train the teacher models in an image-classification manner, using the same data augmentation strategy and loss function as the student detector. Specifically, Faster RCNNs and RetinaNets share the same data augmentation methods, denoted as "general", but use the categorical cross-entropy loss (CEL) and focal loss (FL) for their classification heads, respectively; SSDs have their own data augmentation strategy and use the categorical cross-entropy loss (CEL).

In our experiments, we take ResNet50 as the teacher model. In Section 4.3, we conduct an ablation study with different teachers. Furthermore, we investigate the influence of different input sizes to our classification teachers because the objects in object detection have different resolutions than they typically have in image classification. Therefore, we train the classification teacher with input sizes in $[56 \times 56, 112 \times 112, 224 \times 224]$. Because Faster RCNNs and RetinaNets share the same data augmentation, we train a teacher for both frameworks using the two losses jointly. All the teacher models are trained using ImageNet-pretrained weights for 90 epochs with an initial learning rate of 0.0001, divided by 10 at epoch 50.

---

[1] https://github.com/NVlabs/DICOD
[2] https://cocodataset.org
[3] https://github.com/open-mmlab/mmdetection
[4] https://github.com/open-mmlab/mmcv

35th Conference on Neural Information Processing Systems (NeurIPS 2021).

**Results**. In Table S1, we report the top-1 accuracy of our ResNet50 classification teacher on the COCO2017 classification validation dataset. The teacher models trained with the categorical cross-entropy loss benefit from larger input sizes, as shown by the top-1 accuracy increasing by more than 4 points when the input size increases from 56 to 224. Surprisingly, with the focal loss, increasing the input size to 224 yields slightly worse results than with an input of size 112. Note that the teacher trained with the focal loss underperforms those trained with categorical cross-entropy loss by more than 3 points. Furthermore, training the classification teacher with both losses always yields better top-1 accuracy than training with a single loss. To this end, we will use the same classification teacher for all two-stage Faster RCNNs and one-stage RetinaNets in our classifier-to-detector distillation method. We consider this to be an advantage of our method, since it lets us use the same teacher for multiple detectors.

Table S1: Top-1 accuracy of classification teacher ResNet50 on the COCO2017 classification validation dataset.

| Data Aug. + Loss | Input resolution | | |
|---|---|---|---|
| | $56 \times 56$ | $112 \times 112$ | $224 \times 224$ |
| SSD + CEL | 76.26 | 80.30 | 80.41 |
| general + CEL | 76.92 | 80.81 | 81.42 |
| general + FL | 72.86 | 77.50 | 77.04 |
| general + CEL + FL | **77.01** | **81.02** | **81.67** |

## S3  Training Setting for Compact Students

Let us now specify the details for the training settings of the compact student models used in 4.1, as mentioned in the main paper and answered in Checklist 3(b) and 3(d). All experiments in this work are performed on Tesla V100 GPUs.

**SSD300 and SSD512.** For data augmentation, we first apply photometric distortion transformations on the input image, then scale up the image by a factor chosen randomly between $1\times$ and $4\times$ by filling the blanks with the mean values of the dataset. We then sample a patch from the image so that the minimum IoU with the objects is in $[0.1, 0.3, 0.5, 0.7, 0.9]$, with the precise value chosen randomly. Afterwards, the sampled patch is resized to $300 \times 300$ or $512 \times 512$, normalized by subtracting the mean values of the dataset, and horizontally flipped with a probability of 0.5. We use SGD with an initial learning rate of 0.002 to train the SSDs for 24 epochs, where the dataset is repeated 5 times. The batch size is 64, and the learning rate decays by a factor of 0.1 at the 16th and 22nd epoch.

**Faster RCNN with lightweight backbones.** For data augmentation, the input image is first resized so that either the maximum of the longer side is 1333 pixels, or the maximum of the shorter side is 800 pixels. Then, the image is horizontally flipped with a probability of 0.5. Afterwards, it is normalized by subtracting the mean values and dividing by the standard deviation of the dataset. The Faster RCNN-MobileNetV2 is trained by SGD for 12 epochs with a batch size of 16, and an initial learning rate set to 0.02 and divided by 10 at the 8th and 11th epoch. Faster RCNN-QR50 is trained with a larger batch size of 32 and a larger initial learning rate of 0.04. Note that, in practice, increasing the batch size and the learning rate enables us to shorten the training time while keeping the same performance as with the default $1\times$ training setting in MMDetetion.

## S4  Training longer with $1\times$, $2\times$ and $4\times$ schedulers

To study the effects of longer training on our approach, we trained Faster RCNN-QR50 with $1\times$, $2\times$ and $4\times$ schedulers, and reported the mAP in the Table S2. Our distillation method yields consistent and significant improvements with all training schedulers. This indicates that our method can make the student model converge to a better solution, not just train faster.

Table S2: Results of our classifier-to-detector distillation method with Faster RCNN-QR50 on the COCO2017 validation set for $1\times$, $2\times$ and $4\times$ training schedulers.

| Scheduler | $1\times$ | $2\times$ | $4\times$ |
|---|---|---|---|
| Faster RCNN-QR50 | 23.3 | 23.6 | 24.5 |
| + Ours | **27.2** (↑ **3.9**) | **27.7** (↑ **4.1**) | **28.4** (↑ **3.9**) |

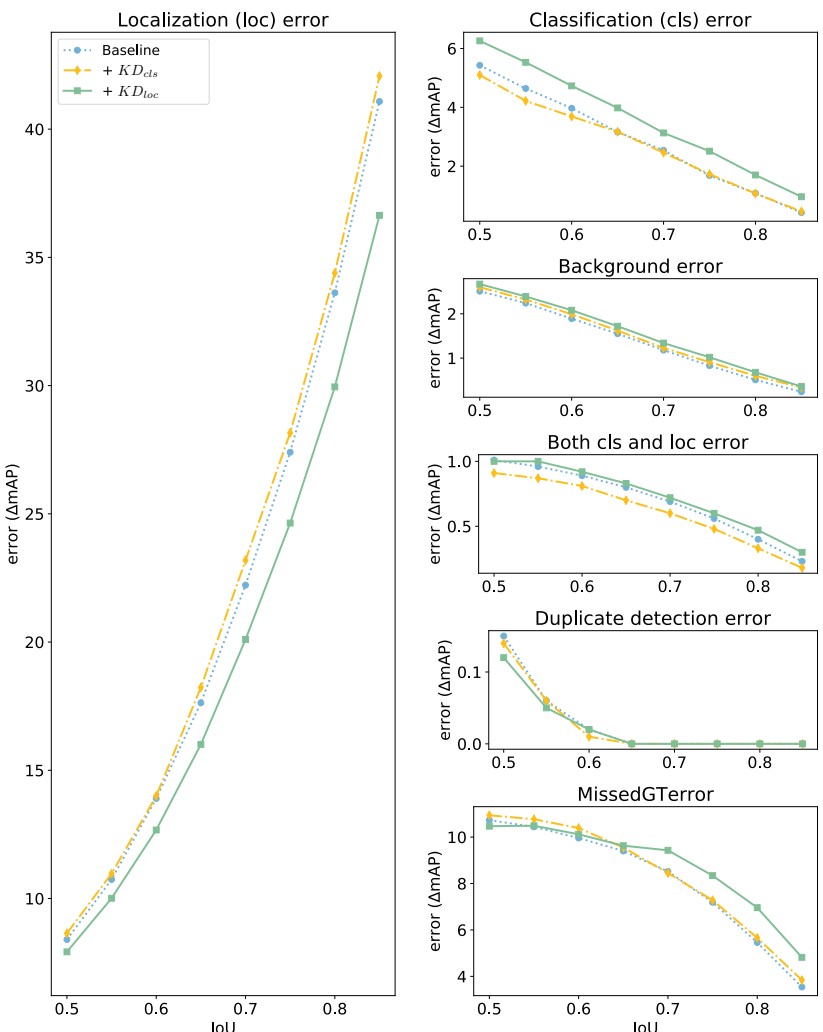

Figure S1: **Detection errors (better viewed in color).** We show 6 types of detection errors for the baseline SSD300, and with our classification and localization distillation methods. Note that we scaled the plots according to the magnitude of the errors they represent; the localization error, classification error and missedGTerror are the main sources of errors.

## S5    Analysis of Detection Errors

As mentioned in Section 4.4 of the main paper, we provide the 6 types of detection errors discussed by Bolya et al. [3], namely, classification (cls) error, localization (loc) error, both cls and loc error, duplicate detection error, background error, missed ground-truth error (missedGTerror).

In essence, as shown by Figure S1, localization error increases significantly as the foreground IoU increases, while all other errors decrease. The classification-related errors typically drop by using our classification distillation strategy. See, for example, the classification error for IoUs smaller than 0.65, and the error of both cls and loc for all IoUs. By contrast, our localization distillation decreases the localization-related errors, including localization error and duplicate detection errors. Specifically, with localization distillation, the localization error drops by more than 2 mAP points for IoUs larger than 0.7, albeit with a marginal increase in missedGTerror and background error. Overall, while there is a tradeoff between our classification and localization distillation strategies, they play complementary roles in improving the performance of the student detector.