# OpenReview forum: "Distilling Image Classifiers in Object Detectors"
_NeurIPS.cc/2021/Conference — NeurIPS 2021 Poster_

### Official Review · Reviewer_twvd · 2021-07-15

**Rating:** 7
**Confidence:** 3

**Summary:**

The paper aims to transfer knowledge from image classifiers to object detectors using distillation. The teacher and student models are trained on the same dataset, and distillation is performed from teacher to student via classification and localization components. The said model improves on the non-distilled baseline student detectors, and also compete with prior related detector-to-detector distillation works. The ablation study demonstrates the effectiveness of the loss terms and also shows the complementary nature of the classification and localization distillation loss terms.


**Limitations And Societal Impact:**

Yes

**Main Review:**

### Strengths

- The claims made in the paper are appropriate and are substantiated by the experimental results and ablation study


### Weaknesses

The problem setup
- Although not exactly a weakness, but the idea of transferring knowledge from image classifiers to object detectors touches upon another line of work where an explicit transformation / mapping function is learned from classifiers to detectors [1, 2], which can then be readily deployed on novel classes for which the detection data is scarce, making the problem highly desirable. However the problem explored in this paper is distillation primarily and obtaining lower memory and compute intensive models (student) that are able to deliver better accuracy. My question is did the authors think about using less detection data and still be able to manage comparable performance to SOTA? That, in my opinion, would be even more impactful.

### Typo

- Line 49: ’be’ —> ‘by’
- Line 215: ’reaches a 2.3 mAP improvement’. Shouldn’t it be 2.6?

### Justification for the rating

Although I’m not super familiar with this line of research, however in my opinion the primary incentive for distillation is to obtain better performing compact students which the paper has empirically demonstrated. I’m leaning towards accept, however I’m keen to hear what the other reviewers might have to say to inform my final rating.


[1] Hoffman, Judy, et al. "LSDA: Large scale detection through adaptation." Advances in neural information processing systems 27 (2014): 3536-3544.

[2] Wang, Yu-Xiong, Deva Ramanan, and Martial Hebert. "Meta-learning to detect rare objects." Proceedings of the IEEE/CVF International Conference on Computer Vision. 2019.


**Time Spent Reviewing:**

6

---

> ### Author Response · Authors · 2021-08-10
> **Response to Reviewer twvd**
>
> Thank you for pointing us to this interesting line of work, which we will discuss in the revised version. [1] proposes an adaptation method to turn a full-class classifier into a detector with a subset of the bounding box annotations, so that the transformed detector is able to predict bounding boxes for new classes whose ground-truth bounding boxes are unavailable. [2] tackles few-shot object detection to detect new object classes. Both papers rely on adaptation or meta layers to learn the meta knowledge from limited data. By contrast, we focus on improving compact detection models trained on a full dataset using a teacher model. We will nonetheless keep in mind the reviewer’s suggestion of using less data for future work.

---

### Official Review · Reviewer_ML2n · 2021-07-16

**Rating:** 4
**Confidence:** 4

**Summary:**

This paper designs a new framework for distillation in object detection tasks. It claimed itself as a new method for opening the cross-task distillation directions. However, it is a bit overstated.

**Main Review:**

This is a nice paper with clear illustrations and structures. I enjoy reading it.

A big issue of this work is that its significance is overstated ("Ultimately, we believe that our work opens the door to a new approach to distillation beyond object detection: Knowledge should be transferred not only across architectures, but also across tasks.") because we can see from table 1, the localization distillation loss almost does not work (mostly only 0.1% mAP improved). What works for the final performance boost is the classification distillation loss which is basically applied between the classifier of the student and the teacher (which is also a classifier) --- the meaningful distillation happens only between the teacher classifier and the student classifier but not between the teacher classifier and student detector. This means that the experimental results in this paper do not support its assumption or motivation.

Other detailed comments:

1. Performance is not impressive compared to related works. From table 2, we can see that "ours" exceeds FKD for 1% when feature plugin is not possible (in below two blocks) and all its other performances (+Ours in the top three blocks) are lower than FKD.

2. Lines 107-109. This is not very interesting actually, as it trains a classifier and distill the knowledge to another classifer (the loc distill not working as mentioned above). While, it would be interesting to see how can apply a normally trained classifier (does not require such a cropping in the images using bounding box labels) for distilling a good detector. The way is to use a pre-trained classifiers (on ImageNet) or a normally trained classifier on target dataset such as multilabel classifiers trained on the weakly supervised detection dataset (i.e., bbox can not be used in training classifiers).

3. Lines 38, 39. It is not significant if distilling is only between classification and detection models as the only arch difference is a regression head in the last layer (output layer). I would be more agreed on the significance if it is from a classification model to a segmentation model where the latter needs a very different arch (an additional multi-layer net to decode a mask).

**Time Spent Reviewing:**

3.5 hours

---

> ### Author Response · Authors · 2021-08-10
> **Response to Reviewer ML2n**
>
> Thank you for your review. We address your comments below. We would appreciate any further feedback.
> ### **Q1: The significance of the paper is overstated.**
> ***We respectfully disagree with the reviewer and believe that this comment stems from their misunderstanding of some key concepts in object detection and in our distillation strategies. We clarify these points below and hope the reviewer will reconsider their score.***
>
> First, we introduce the concept of cross-task distillation, studying in detail the case of using an image classifier as a teacher for an object detector. Our results demonstrate the effectiveness of such a classifier-to-detector distillation, and evidence that a classification teacher brings new knowledge that is complementary to that obtained from a detection teacher. This has never been studied before and, we believe, is of interest to the community.
>
> Furthermore, and as clarified in more detail below, our localization distillation indeed works, and our classification distillation is not as naive as the reviewer may have initially thought.
>
> >“...the localization distillation loss almost does not work...”
>
> As discussed in detail in our answer to Reviewer q5n7 (Q4, Q5), our localization distillation consists of two parts: $loc$ (Eq. 11) and $KD_{loc}$ (Eq. 10), which are both our contributions. In fact, $L_{loc}$ in Eq. 11 can be thought of as a special case of the localization distillation of Eq. 10, making it efficient even without a teacher. In short, $L_{loc}$ is less strict than a standard regression loss because it links the regressed bounding boxes to lower resolution feature maps. Altogether, our localization distillation method ($loc$ + $KD_{loc}$ in Table 1) improves the baseline by 1.6, 1.6, 1.0 and 0,4 mAP points with std < 0.1. Moreover, as shown in Table4(c), when using them separately ($l_0$ corresponds to ‘$KD_{loc}$’ and $obj$ to $‘loc$’), they each outperform the baseline by more than 0.5 points. These numbers evidence the significance of our localization distillation approach. We will clarify this.
>
> >“ ..the meaningful distillation happens only between the teacher classifier and the student classifier but not between the teacher classifier and student detector...”
>
> As discussed above, this is incorrect, as our distillation strategy also improves the localization of the bounding boxes. Furthermore, as discussed in our answer to Reviewer q5n7 (Q2), the classifier in an object detector is not the same as a standard image classifier.  We show the main differences between them below:
>
> |    	| Input                               	| Output        	| Target       | Loss     |
> |---------------------------	|---------------------------------------------------------------------------	|------------------------------------------------	|-------------------------------------------------------	|-----------	|
> | standard image classifier 	| Images with one centered object                                           	| foreground classes distribution                	| one-hot label for each image                          	|  CE       	|
> | classifier in detector    	| Images with multiple objects from multiple classes and at different scales.  Each class may have multiple instances. 	| foreground and background classes distribution 	| Multiple one-hot labels for all objects in each image 	| CE or BCE 	|
> |                           	|                                   	|                                                	|                                                       	|           	|
>
> Because of these differences, it is impossible to directly distill knowledge from a standard image classifier to the classifier in the detector. To address this, we
> - derived a KL-divergence-based loss function for logits employed in binary cross-entropy losses.
> - tackled the case where the classes differ between the teacher and the student, because the teacher focuses on the foreground classes only.
>
> We consider this as a part of our contributions.
>
> Note also that, in contrast to detector-to-detector distillation methods, which rely on different teachers for one-stage and two-stage detectors, our approach allows us to train a ***single*** general teacher applicable to multiple detector types.  In our experiments, we used the same classification teacher for all two-stage detectors and for the one-stage RetinaNet. We only used a different teacher for SSD detectors because they use different data augmentation strategies. We see the generality of our classification teacher as another strength of our approach.
> ### **Q2: “Performance is not impressive compared to related works”.**
> As shown in Table 2,  for Faster RCNN-QR50 and SSD512, the state-of-the-art detector-to-detector distillation approach, FKD, improves the baseline by 2.8 and 1.8 mAP, while our method outperforms FKD by 1.1 mAP and 0.9 mAP, respectively. This corresponds to almost ***50%*** of improvement over FKD.
>
> For Faster RCNN-R50 and RetinaNet-R50, FKD outperforms its competitors by 1.3 mAP and 0.5 mAP, and our approach further boosts FKD by 0.4 (+***31%*** improvement), and 1.1 (+over ***200%*** improvement), respectively.
>
> Overall, our approach is orthogonal to the detector-to-detector distillation methods, allowing us to achieve state-of-the-art performance by itself or by combining it with the detector-to-detector distillation strategy.
>
> ### **Q3: The classification distillation is not interesting.**
> > “ ...as it trains a classifier and distill the knowledge to another classifer… ”
>
> As discussed in Q1, our approach to classification distillation is not as straightforward as Reviewer ML2n believes. While, conceptually, it still distills knowledge to another classifier, the differences between the two classifiers are such that this would have been impossible without our contributions.
>
> ### **Q4: Distilling from normally trained classifier teachers.**
> > “it would be interesting to see how can apply a normally trained classifier ... for distilling a good detector...”
>
> Thanks for the suggestion to use an ImageNet pre-trained classifier or an image-level multilabel classifier. We implemented the first idea by using a ResNet50 pre-trained on ImageNet and replacing our classification teacher with it. For the second idea, we trained an image-level multilabel classifier ResNet50 on MS COCO. In this case, we then used a distillation loss similar to that of Eq. 1 to compare the distribution obtained with the multilabel teacher to that obtained from the student detector by grouping its positive class predictions. We compare the results of our KD$_{cls}$ approach with different classifier teachers for Faster RCNN-QR50 in the following table.
>
> |      $\ \ \ \ \ \ \ \  $      Teacher  	|    mAP   	|   AP$_s$ |   AP$_m$   |   AP$_l $   	|
> |---------------------:	|:--------:	|:--------:	|:--------:	|:--------:	|
> |            Baseline (w/o teacher) 	|   23.3   	|   13.1   	|   25.0   	|   30.7   	|
> | ImageNet pre-trained  |   21.0   	|   11.6   	|   22.9   	|   27.6   	|
> |        Multilabel  |   23.5   	|   13.0   	|   25.5   	|   30.8   	|
> |                 Ours | **25.9** 	| **15.3** 	| **27.9** 	| **34.0** 	|
> | | | | |
>
> Not surprisingly, the ImageNet pre-trained classifier performs poorly because it has no knowledge of class distributions over the COCO dataset. The image-level multilabel classifier teacher slightly improves the baseline by 0.2 mAP points, which remains 2.4 mAP points worse than our classifier teacher.
>
> These experiments further evidence the benefits of our classification teacher and distillation approach. We will incorporate them in the revised paper.
>
> ### **Q5: The only arch difference between a classifier and a detector is a regression head in the last layer (output layer).**
> We respectfully disagree with the statement. Specifically:
> - Classification and detection frameworks differ much more fundamentally.
>
> For example, as shown in Figures 2 and 3 of the Faster RCNN paper [33], a detector architecture contains carefully designed modules, such as the region proposal network (RPN) to select bounding boxes. Even for simpler, one-stage detectors, such as YoloV3, the modules after the backbone contain much more than a regression head in the last output layer. See, e.g., [r1] for more details.
> - We use different architectures for our classification teacher and for the student detectors.
>
> In our experiments, we use a ResNet50 as our classification teacher, while the backbones of the student detectors include a VGG16, a ResNet50 with a quarter of the channels, and a MobileNetV2.
>
> We therefore believe that our classifier-to-detector distillation constitutes a valid example of cross-task distillation and will be valuable to the community.
>
> [r1] YoloV3 architecture, http://media5.datahacker.rs/2020/03/11-1024x423-new.jpg
>
> ### **Q6: “...I would be more agreed on the significance if it is from a classification model to a segmentation model...”**
> Although detection models use some classification losses, classification and detection are two fundamentally different tasks. In short, classification discards the spatial information from the input, whereas detection needs to preserve such spatial information during the forward pass, making it a much more challenging task than classification. By contrast, segmentation networks perform a pixel-wise classification, which lets them use the same classification loss as in standard classification.
>
> Therefore, we believe it to be fair to say that the difference between classification and detection is no less than between classification and segmentation, and that our work will thus be valuable to the community.
>
> We nonetheless acknowledge that distilling a classifier in a segmentation student would be an interesting direction to further study cross-task distillation, but it goes beyond the scope of this paper.

---

> > ### Comment · Reviewer_ML2n · 2021-09-02
> > **final decision**
> >
> > Thanks for the responses.
> >
> > My key concerns are still there.
> >
> > 1. For “...the localization distillation loss almost does not work...”
> >
> > I meant that "+loc+KD_loc" has no clear improvement (0.1%) over "+loc", so there is no evidence of "the significance of the localization distillation approach" (the "contribution" over the standard classifier-to-classifier distillation is too limited).
> >
> > 2. "they each outperform the baseline by more than 0.5 points"
> >
> > This seems a claim related to table 4c, but I can not find the baseline number 26.7 comparable to the best number 27.2 (as 27.2-0.5 = 26.7). A number 26.7 is shown in 4a, but it is not shared as a baseline in the other two tables 4b and 4c. Not sure if I am wrongly navigated.
> >
> > 3. "As shown in Table 2, for Faster RCNN-QR50 and SSD512, the state-of-the-art detector-to-detector distillation approach, FKD, improves the baseline by 2.8 and 1.8 mAP, while our method outperforms FKD by 1.1 mAP and 0.9 mAP, respectively. This corresponds to almost 50% of improvement over FKD.
> > For Faster RCNN-R50 and RetinaNet-R50, FKD outperforms its competitors by 1.3 mAP and 0.5 mAP, and our approach further boosts FKD by 0.4 (+31% improvement), and 1.1 (+over 200% improvement), respectively."
> >
> > It seems the authors try to avoid discussing or explaining the reasons for the low performance of +Ours over +FKD on two stronger networks: Faster RCNN-R50 and RetinaNet-R50 ("stronger" because of their higher performance than the other three backbones/detectors in the overall performance e.g. mAP). Not convinced to me. If the method performs lower on stronger backbones, the method is weaker. In other words, the conclusion is +Ours is not comparable to +FKD when the backbone is good enough. In fact, based on deep learning, the backbone will become better and better, so this method might not be chosen (if there is another choise called FKD).
> >
> > 4. "We implemented the first idea by using a ResNet50 pre-trained on ImageNet and replacing our classification teacher with it. For the second idea, we trained an image-level multilabel classifier ResNet50 on MS COCO. In this case, we then used a distillation loss similar to that of Eq. 1 to compare the distribution obtained with the multilabel teacher to that obtained from the student detector by grouping its positive class predictions. We compare the results of our KD approach with different classifier teachers for Faster RCNN-QR50 in the following table."
> >
> > I would have been convinced if the method really makes stronger distillation between different tasks, i.e., from normal classification models to advanced detection models. However, as shown in the supplement table ("the following table"), +Ours using the images cropped by bbox (the annotation given in detection datasets) always performs the best, and it NEEDs detection dataset (multi objects and precise bbox annotated). In other words, distilling from a low-level model (classification) to solve a high-level task (detection) is still not feasible. The performance is limited due to the poor localization knowledge in the low-level teacher (the classification model trained on normal single-label images is poor to transit any effective localization knowledge to the training of a detection model, which is a conflict with the motivation in this paper).
> >
> > Overall, I am not convinced regarding the poor performance of the method and the vague motivation of this submission. I will keep my rating.

---

> > > ### Author Response · Authors · 2021-09-02
> > > **Thank you for your feedback.**
> > >
> > > **1**: We would like to clarify again: **BOTH** ‘+loc’ and ‘+loc+kd_loc’ are our contributions. ‘+loc’ performs well, and ‘+loc+kd_loc’ further boosts the results.
> > >
> > > **2**: *"they each outperform the baseline by more than 0.5 points".*
> > >
> > > By “they”, we meant all the numbers in Table 4(c), and the ‘baseline’ is the ‘baseline’ in Table 3(a) yielding an mAP of 26.3, which is shared across Table 3 and Table 4 for the ablation study. All the 4 numbers (27.1, 26.8, 27.2, 26.9) in Table 4(c) show an improvement of more than 0.5 points using our localization distillation method.
> > >
> > > **3**: We respectfully disagree with this comment because we explicitly show and discuss the results of +Ours and +FKD for the two stronger detectors in our paper (line 228 to line 244, Table 2) and in the first rebuttal. The goal of knowledge distillation is to improve small models, for example for embedded platforms. Our method yields better performance than FKD for smaller models, and for deeper ones, it provides a further boost to FKD. This makes our approach valuable; The **BEST** choice is to use **Ours + FKD**.
> > >
> > > **4**:
> > >
> > > a. *“it NEEDs detection dataset (multi objects and precise bbox annotated)...distilling from a low-level model (classification) to solve a high-level task (detection) is still not feasible”.*
> > >
> > > Prior to our work, distillation for object detection has been addressed solely from a detection teacher to a detection student. Our work performs distillation from a classification teacher to a detection student. Both networks are trained using the same dataset, as is common practice in knowledge distillation. Nevertheless, our approach leverages a different kind of knowledge coming from our classification teacher, which facilitates the training of the student detector. In other words, our method and results show that distilling from a classifier to a detector is feasible.
> > >
> > > b. *“The performance is limited due to the poor localization...which is a conflict with the motivation in this paper”.*
> > >
> > > Our classification and localization distillations do not happen at the image level but at the bounding box (object) one based on the properties of the detection pipeline. Thus, while “the classification model trained on normal single-label images” indeed performs poorly, **OUR** classification model trained on cropped objects is effective for distillation, including for localization.
> > >
> > > We hope our responses clarify the reviewer’s concerns.

---

> > > > ### Comment · Reviewer_ML2n · 2021-09-03
> > > > **feedbacks**
> > > >
> > > > 1. +loc is just a normal term used in the distillation between detectors (student and teacher detectors). If the one used in this paper is slightly different from the normal one, this does not make a significant novelty to the approach. The main improvement so far relies too much on this normal term rather than the proposed (more novel one) term KD_loc, which is a bit disappointing.
> > > >
> > > > 2. thanks for the clarification of where the baseline is.
> > > >
> > > > 3. Given the same complexity of using a plugin module, one may choose to use +FKD on strong backbones. This is what I meant.
> > > >
> > > > 4. Overall, I think the key problem is the conflict between good motivation (in the beginning) and disappointing results (in the end). It will be more valuable if the high-level (or more data-hungry) models can be effectively distilled or learned from a low-level one (e.g. normally trained classification models). It will be a new way of knowledge transfer between models (a replacement of transfer learning using pre-trained weights as initialization). However, as shown in the paper's and supplemented results, using normal classification models does not help too much for distilling a good detector. Relatively good results can only be achieved by using the same annotations (bbox + object labels) as those used for training detectors. Given the full detection dataset, why not directly distill between two detectors or develop some advanced distillation between two detectors. What is the motivation for using the classification model, then?
> > > >
> > > > Thanks.

---

> > > > > ### Author Response · Authors · 2021-09-03
> > > > > **Thank you for the further feedback.**
> > > > >
> > > > > 1.  ‘+loc ‘ is **NOT** ’between detectors (student and teacher detectors)‘ but proposed by us between our classification teacher and the detection student. As mentioned before and stated at lines 174-181 of the paper, ‘+loc’ is **indeed** our contribution and a special case of ‘KD_loc’.
> > > > >
> > > > >     As stated in answer to Q4 of Reviewer q5n7:
> > > > >     > ‘+loc’ is general, novel and it is a part of our contributions. In fact, $L_{loc}$ in Eq. 11 can be considered as a special case of the localization distillation of Eq. 10, making it efficient even without a teacher. In short, $L_{loc}$ is less strict than a standard regression loss because it links the regressed bounding boxes to lower resolution feature maps. We will emphasize that $L_{loc}$ is also our contribution and clarify its discussion.
> > > > >
> > > > > 3\.  We respectfully cannot agree with the reviewer on this point. The main goal of knowledge distillation is to train a better compact student network. To this end, one uses extra knowledge during training, while keeping a low computational cost **at inference**. In short, for a given student model of fixed complexity **at inference time**, the best way to achieve the SOTA results is to use **+Ours + FKD**.
> > > > >
> > > > > 4.
> > > > >
> > > > > - a. *“It will be more valuable if the high-level (or more data-hungry) models can be effectively distilled or learned from a low-level one (e.g. normally trained classification models). It will be a new way of knowledge transfer between models (a replacement of transfer learning using pre-trained weights as initialization).”*
> > > > >
> > > > >     We tackle neither the weakly-supervised object detection scenario nor the transfer learning one, and never claimed to do so. We address knowledge distillation for object detection. Our classification teacher is trained **normally**, but based on the objects and their labels from the detection dataset (details of training our classification teacher are provided in the supplementary material, section S2.). Therefore, we perform distillation between models that were trained for different tasks (albeit the same data), which brings a different kind of knowledge in the student model than using a detection teacher.
> > > > >
> > > > >
> > > > > - b. *”Given the full detection dataset, why not directly distill between two detectors or develop some advanced distillation between two detectors. What is the motivation for using the classification model, then?”*
> > > > >
> > > > >     As stated in answer to Q1 of Reviewer q5n7:
> > > > >
> > > > >     > Our approach was motivated by our observation that the classification head of a detector, even a deep one, yields inferior performance compared to what can be expected from a pure image classifier. Even if our classification teacher is trained on the same dataset as what would be used for an integrated detection teacher, it tackles a different task and is trained in a different manner. Therefore, our classification teacher provides a different knowledge to the student, for both classification and localization, than that extracted by an integrated detection teacher.
> > > > >
> > > > >     > The benefits of our cross-task distillation approach, which had never been studied before, are evidenced by our results. Our classifier-to-detector strategy yields state-of-the-art results for compact detectors and is complementary to detector-to-detector distillation thus further boosting its performance. Furthermore, in contrast to detector-to-detector distillation methods, which rely on different teachers for one-stage and two-stage detectors, our approach allows us to train a single general classification teacher applicable to multiple detector types (details of training our classification teacher are provided in the supplementary material, section S2.).
> > > > >
> > > > > In short, distilling between two detectors is not perfect and our approach provides an alternative, effective on its own and complementary to detector-to-detector distillation. This is acknowledged by both Reviewers q5n7 and twvd. We will clarify and highlight our motivation in the revised version.
> > > > >
> > > > > Thank you for the discussion.

---

### Official Review · Reviewer_q5n7 · 2021-07-18

**Rating:** 5
**Confidence:** 4

**Summary:**

The paper proposes a method for distilling a low-compute object detector. Contrary to prior work, which used a more standard object detector-to-object detector distillation approach (where object detectors were most often restricted to be of the same category), the authors propose to use a classification backbone as teacher. To do this, the classifier is trained on the ground truth bounding boxes, and then used to impose both a classification head distillation, and a localization head distillation. The classification distillation is rather straightforward (using a logit matching approach), while the localization distillation is more involved. A spatial transformer is used on top of the bounding box prediction, and a feature matching loss is imposed so that the predicted bbox -> transfomer -> adaptive pooling branch, and the ground truth bbox -> classifier teacher -> adaptive pooling branch are similar to each other.

**Limitations And Societal Impact:**

The authors claim that they discuss the limitations on Section 5. This is not really the case, no such discussion actually happens.

Societal impact is discussed... although it is more of a variant of the conclusions. I am however of the opinion that no specific negative societal impact should be highlighted for this work as it is a rather generic approach with applicability to any object detection method.

**Main Review:**

The premise of the paper is clear, and proposes a scenario that is novel (classifier to detector distillation). The proposed scenario might also have some interesting practical implications, though this is not really discussed much in the paper. In the end, the classifier is trained on the dataset itself, why is this better than using an integrated localization and classifier model? I understand being able to use a large-scale classification dataset would be an advantage, but why does it work better given that the method trains on the detection dataset?

The technical approach is split into two: the classifier distillation itself and the localization distillation (which uses two losses). The first bit is very standard and it does not have any novelty (classifer-to-detector scenario aside). The second is more novel, but I have questions about it (see comments bellow).

The paper is mostly well written, but in my opinion would benefit from a re-balancing of what the limited space is devoted to: the details of the classifier distillation are very standard and in some cases could be omitted altogether (is ok to leave softmax computation out, even with temperature, eq. 5 and 6 could occupy less space - particularly 6 could be removed, etc.). Instead, the localization distillation is much more vague. For example, the spatial transformer bit will not make the idea clear to someone that is not familiar with the spatial transformer, and the adaptive pooling is simply not introduced at all - and this is far from a standard technique.

Technically speaking, the method mixes simple concepts with some that are more obscure to me. The localization distillation is very unclear actually. The spatial transformer is introduced in-between the output of the localization of the student network, and the second bit of the distillation strategy, that compares the output of the spatial transformer to the classification feature. But what confuses me the most is, unless you actually keep the spatial transformer at test time on top of the bounding box prediction (like an alternative bounding box regression), then the only way it can influence the parameters involved at test time is very limited. This had me quite confused during section 3.2, thinking I misunderstood what the authors were proposing, but then I saw that the impact is very limited (ablation study on Table 1). Instead, the localization branch learns through the second localization loss (termed "loc" in the paper), which does not involve a teacher at all. So, to sum up, I'm not sure the contributions as claimed are the relevant bits, and I'm a bit confused of why a term that does not involve a teacher at all has such positive effect - this sounds like a general-purpose module, is it novel?

In terms of results, Ours + FKD offers good improvements over "Ours". Why is it not shown for every student network considered?

I'm open to upgrading the score if the rebuttal clarifies the methodological doubts I expressed above.

It would be nice to see the results when using 1x, 2x, 4x scheduler. KD can make models train faster, but maybe not better if trained for long enough?


Small issues (no need to address them on rebuttal):

"Heo et al. [17] provides a comprehensive study of feature distillation." This is not literature review, it is taking the title of the paper and making it a phrase. No insight provided.

Some details on how the classifier is trained would be welcome. e.g. the heuristics of patch cropping, augmentations, schedule, hyperparameters, etc

References need a bit more care. For example, Sec. 4.1 the reference provided for ImageNet has nothing to do with the dataset creation (it is "just" a method to classify on imagenet). SSD and Faster RCNN are introduced before but it would not be bad to give the reference here too, but MobileNetV2 and QR50 do not have a reference (not sure where the QR50 comes from either). And for example it's MMDetection, not mmdetection.



**Time Spent Reviewing:**

3

---

> ### Author Response · Authors · 2021-08-10
> **Response to Reviewer q5n7**
>
>
> ***Thank you for the detailed and valuable reviews. We hope that our explanations address the reviewer’s concerns so as to make them upgrade their score. We would appreciate any further feedback and discussions.***
>
> ### **Q1: Why is a classification teacher better than an integrated detection teacher?**
>
> Our approach was motivated by our observation that the classification head of a detector, even a deep one, yields inferior performance compared to what can be expected from a pure image classifier. Even if our classification teacher is trained on the same dataset as what would be used for an integrated detection teacher, it tackles a different task and is trained in a different manner. Therefore, our classification teacher provides a different knowledge to the student, for both classification and localization, than that extracted by an integrated detection teacher.
>
> The benefits of our cross-task distillation approach, which had never been studied before, are evidenced by our results. Our classifier-to-detector strategy yields state-of-the-art results for compact detectors and is complementary to detector-to-detector distillation thus further boosting its performance. Furthermore, in contrast to detector-to-detector distillation methods, which rely on different teachers for one-stage and two-stage detectors, our approach allows us to train a ***single*** general classification teacher applicable to multiple detector types (details of training our classification teacher are provided in the supplementary material, section S2.).
>
> ### **Q2: The classifier distillation is very standard and does not have any novelty.**
>
> While we acknowledge that classifier distillation has often been used in the literature, we respectfully disagree with the reviewer’s statement. Our method is the ***first*** to use a classification teacher to distill knowledge to student detectors, thus introducing the fundamentally novel problem of cross-task distillation. We believe this to be of interest to the community.
>
> Furthermore, to tackle the classifier-to-detector distillation scenario, we
> - derive a KL-divergence-based loss function for detection models employing binary cross-entropy losses;
> - address the case where the classes differ between the teacher and the student, because the teacher focuses on the foreground classes only;
> - introduce an approach to exploiting the classification teacher to distill localization knowledge in the student detector.
>
> As suggested by the reviewer, we will move some of the classification distillation details to the supplementary material and provide more detail on localization distillation so as to make it more accessible to the reader.
>
>
> ### **Q3: Is the spatial transformer used at test time?**
> No, it is not involved at test time. Because no distillation occurs at test time, there is no need to crop the regressed bounding boxes anymore and thus no need for the spatial transformer.
>
> To clarify, the spatial transformer is a non-parametric differentiable module that links the regressed bounding boxes with the classification teacher to yield an end-to-end model during training. This allows the localization distillation loss to contribute to updating the parameters of the student detector, aiming to improve the regressed bounding boxes. We will clarify this.
>
>
> ### **Q4: Is the localization distillation a general module and is it novel?**
>
> Yes, it is general, novel and it is a part of our contributions. In fact, $L_{loc}$ in Eq. 11 can be considered as a special case of the localization distillation of Eq. 10, making it efficient even without a teacher. In short, $L_{loc}$ is less strict than a standard regression loss because it links the regressed bounding boxes to lower resolution feature maps. We will emphasize that $L_{loc}$ is also our contribution and clarify its discussion.
>
>
> ### **Q5: The impact of $KD_{loc}$ is limited.**
>
> > “...the impact is very limited (ablation study on Table 1)”
>
> The results in Table 1 were obtained using $KD_{loc}$ and $loc$ jointly. The benefits of  $KD_{loc}$ on its own can be seen from Table4(c). In short, by itself, $KD_{loc}$ (which corresponds to $l_0$ in Table 4(c)), improves the performance from 26.3 to 26.8. Using it together with $loc$ ($obj$ in Table 4(c)) yields the best result of 27.2. We acknowledge that this was not clear from the table and will improve this discussion.
>
> ### **Q6: More experiment results.**
> - Ours+FKD for Faster-RCNN-QR50 and SSD512.
>
> As shown in Table 2, for Faster-RCNN-QR50 and SSD512, our classifier-to-detector approach yields better results than FKD by using our classification teacher only. We would nonetheless be happy to include the Ours+FKD results for them.
>
> As shown below, Ours+FKD achieves SOTA results, which further evidences the effectiveness of our classifier-to-detector distillation approach and its complementarity to detector-to-detector distillation.
>
> |      Method      	|  mAP 	|
> |---------------:	|:----:	|
> | Faster RCNN-QR50 	| 23.3 	|
> |            + FKD 	| 26.1 	|
> |           + Ours 	| 27.2 	|
> |     + Ours + FKD 	| **28.0** 	|
> |                  	|      	|
> | SSD512-VGG16     	| 29.4 	|
> |            + FKD 	| 31.2 	|
> |           + Ours 	| 32.1 	|
> |     + Ours + FKD 	| **32.6** 	|
> |                  	|      	|
>
>
> - Results when using 1x, 2x, 4x scheduler
>
> Following the reviewer’s suggestion, we trained Faster RCNN-QR50 with 1x, 2x, 4x schedulers, and reported the mAP in the following table. Our distillation method yields consistent and significant improvements with all training schedules. This indicates that our method can make the student model converge to a better solution, not just train faster.
>
> |      Method      |        $\ \ \ \ \  $       1x     	|     $ \ \ \ \  \  $     2x     	|      $\ \ \ \ \  $    4x     	|
> |----------------:|:----------:	|:-----------:	|:-----------:	|
> | Faster RCNN-QR50 |    23.3    	|     23.6    	|     24.5    	|
> |           + Ours | 27.2 (**+3.9**)	| 27.7 (**+4.1**) 	| 28.4 (**+3.9**) 	|
> |                  |      	| |
>
> ### **Q7: Other suggestions.**
>
> - Details of training the classifier teacher.
>
> We provide the details in the supplementary material, Section S2.
> - References.
>
> Thank you. We will make sure to check and revise the references.

---

> > ### Comment · Reviewer_q5n7 · 2021-08-20
> > **Thanks for the extensive rebuttal**
> >
> >
> >
> > Q1: I think the scenario is interesting but maybe not the most interesting (maybe it is the simplest?). Classification data is easy(er) to gather and annotate, and object proposals are agnostic. Do I need to annotate bounding boxes to train a detector to recognize specific classes? can I leverage the huge size of pre-existing classification datasets to improve the classifier of a detector? etc. I know this is out of scope, so just a discussion for the future.
> >
> > The most important point for the rebuttal in my opinion is:
> >
> > Q3: This is the part where I'm struggling to get my head around. If the Transformer is dumped at test time, then you are allowing the transformer to learn to localize better during training, and then dumping it at test time?
> > The spatial transformer IS parametric, and the parameters of the transformation are computed based on the input tensor. So, a spatial transformer can "correct" your alignment. If the spatial transformer shoulders that role, then the original bounding box does not need to be improved. Maybe there is some terminology issue somewhere?
> >
> > ----------
> >
> > Besides:
> >
> > Q2: Maybe you can include the full phrase you use as the question title: "The first bit is very standard and it does not have any novelty (classifier-to-detector scenario aside)". I hope it is clear that in here I'm referring to the technique, so aside from the novelty of the scenario, which has been recognized on other parts of the rebuttal. And I think it is a fair comment... technically speaking, it is the straightforward thing to do
> >
> > Q6: I think adding the two results would make it look better - it is always nice to see that two techniques would stack up
> >
> >
> > by the way, if I download the paper's pdf and open in using acrobat, image 1 isn't rendered and I get some error message. I can however open the pdf and see image 1 on chrome... it is almost certainly a local issue but just in case

---

> > > ### Author Response · Authors · 2021-08-23
> > > **Thank you for your valuable feedback!**
> > >
> > > **Q1**: Thank you for the comment. In this work, we focus on exploiting the existing annotations of detection datasets to improve the performance of a student detection network. We nonetheless acknowledge the value of the proposed directions and will discuss them as avenues to further investigate our cross-task distillation scenario.
> > >
> > > **Q3**: Yes, there seems to be a terminology issue. In the “spatial transformer networks” paper (STN), the "grid generator" is formulated by $\theta$, which is output by the localization network that processes the input tensor. In other words, a complete STN contains a localization module and a transformation one. In our work, we only use the transformation module, which is non-parametric (i.e. no trainable parameters), and treat the detector as the localization module that generates $\theta$ (in our paper it's $A_k$ in Eq. 8) via the regressed bounding boxes. Our localization distillation improves the resulting $\theta$, and thus the regressed bounding boxes. At test time, we can then remove the transformation module, which is only used to compute our distillation loss during training, without removing trained network parameters. We hope that this clarifies the reviewer’s concern and will revise the text accordingly.
> > >
> > > **Q2**:  Thank you for the clarification. Then, yes, we can agree on this point with the reviewer, but believe that this was nonetheless worth investigating in our cross-task distillation scenario.
> > >
> > > **Q6**: We agree and will definitely add these results to the final version. Thanks!
> > >
> > > **Figure 1**: Thanks for letting us know! We will make sure that the figure also appears in acrobat.

---

### Decision · Program_Chairs · 2021-09-27

**Decision:**

Accept (Poster)

**Comment:**

Two of the reviewers agreed that the paper is novel and can result in cheaper, higher-performing object detectors. The third reviewer misunderstood the term "spatial transformer" and was confused by its use, not responding to the authors' clarification. Given the strong experimental result and interesting approach, I support the acceptance of this paper.